# Program Synthesis with Pragmatic Communication

**Yewen Pu**
MIT

**Kevin Ellis**\*
MIT

**Marta Kryven**\*
MIT

**Joshua B. Tenenbaum**
MIT

**Armando Solar-Lezama**
MIT

## Abstract

Program synthesis techniques construct or infer programs from user-provided specifications, such as input-output examples. Yet most specifications, especially those given by end-users, leave the synthesis problem radically ill-posed, because many programs may simultaneously satisfy the specification. Prior work resolves this ambiguity by using various inductive biases, such as a preference for simpler programs. This work introduces a new inductive bias derived by modeling the program synthesis task as rational communication, drawing insights from recursive reasoning models of pragmatics. Given a specification, we score a candidate program both on its consistency with the specification, and also whether a rational speaker would chose this particular specification to communicate that program. We develop efficient algorithms for such an approach when learning from input-output examples, and build a pragmatic program synthesizer over a simple grid-like layout domain. A user study finds that end-user participants communicate more effectively with the pragmatic program synthesizer over a non-pragmatic one.

## 1 Introduction

Programming is a frustrating process: as the computer executes your code literally, any error in communicating **how** the computer should run would result in a bug. Program synthesis [1] aims to address this problem by allowing the user to specify **what** the program should do; provided this specification, a program synthesizer infers a program that satisfies it. One of the most well-known program synthesizers is FlashFill [2], which synthesizes string transformations from input/output examples. For instance, "Gordon Freeman" $\rightarrow$ "G", the FlashFill system infers the program: "first_letter(first_word(input))". FlashFill works inside Microsoft Excel, and this program can then run on the rest of the spreadsheet, saving time for end-users. However, most specifications, especially those provided by a naive end-user, leave the synthesis problem ill-posed as there may be many programs that satisfy the specification. Here we introduce a new paradigm for resolving this ambiguity. We think of program synthesis as a kind of communication between the user and the synthesizer. Framed as communication we can deploy ideas from computational linguistics, namely *pragmatics*, the study of how informative speakers select their utterances, and how astute listeners infer intent from these "pragmatic" utterances [3]. Intuitively, a pragmatic program synthesizer goes beyond the literal meaning of the specification, and asks **why** an informative user would select that specification.

Resolving the ambiguity inherent in program synthesis has received much attention. Broadly, prior work imposes some form of inductive bias over the space of programs. In a program synthesizer without any built-in inductive bias [1], given a specification $D$, the synthesizer might return any program consistent with $D$. Interacting with such a synthesizer runs the risk of getting an unintuitive program that is only "technically correct". For instance, given an example "Richard Feynman" $\rightarrow$

---

"Mr Feynman", the synthesizer might output a program that prints "Mr Feynman" verbatim on all inputs. Systems such as [4] introduce a notion of syntactic naturalness in the form a prior over the set of programs: $P(prog|D) \propto \mathbb{1}[prog \vdash D] P_\theta(prog)$, where $prog \vdash D$ means $prog$ is consistent with spec $D$, and $P_\theta(prog)$ is a prior with parameters $\theta$. For instance $P_\theta$ might disprefer constant strings. However, purely syntactic priors can be insufficient: the FlashFill-like system in [5] penalizes constant strings, making its synthesizer explain the "r" in "Mr Feynman" with the "r" from "Richard"; when the program synthesized from "Richard Feynman"→"Mr Feynman" executes on "Stephen Wolfram", it outputs "Ms Wolfram." This failure in part motivated the work in [6], which addresses failure such as these via handcrafted features. In this work we take a step back and ask: what are the general principles of communication from which these patterns of inductive reasoning could emerge?

We will present a qualitatively different inductive bias, drawing insights from probabilistic recursive reasoning models of pragmatics [7]. Confronted with a set of programs all satisfying the specification, the synthesizer asks the question, "*why* would a pragmatic speaker use this particular specification to communicate that program?" Mathematically our model works as follows. First, we model a synthesizer without any inductive bias as a *literal listener* $L_0$: $P_{L_0}(prog|D) \propto \mathbb{1}[prog \vdash D]$. Second, we model a *pragmatic speaker*, which is a conditional distribution over specifications, $S_1$: $P_{S_1}(D|prog) \propto P_{L_0}(prog|D)$. This "speaker" generates a specification $D$ in proportion to the probability $L_0$ would recover the program $prog$ given $D$. Last, we obtain the *pragmatic listener*, $L_1$: $P_{L_1}(prog|D) \propto P_{S_1}(D|prog)$, which is the synthesizer with the desirable inductive bias. It is worth noting that the inductive biases present in $L_1$ are *derived* from first principles of communication and the synthesis task, rather than trained on actual data of end-user interactions.

Algorithmically, computing these probabilities is challenging because they are given as unnormalized proportionalities. Specifically, $P_{L_0}$ requires summing over the set of consistent programs given $D$, and $P_{S_1}$ requires summing over the set of all possible specifications given $prog$. To this end, rather than tackling the difficult problem of *searching* for a correct program given a specification, a challenging research field in its own right [8–16], we work over a small enough domain such that the search problem can be efficiently solved with a simple version space algebra [17]. We develop an efficient inference algorithm to compute these probabilities exactly, and then build a working program synthesizer with these inference algorithms. In conducting a user study on Amazon Mechanical Turk, we find that naive end-users communicate more efficiently with a pragmatic program synthesizer compared to its literal variant. Concretely, this work makes the following contributions:

1. a systematic formulation of recursive pragmatics within program synthesis
2. an efficient implementation of an incremental pragmatic model via version space algebra
3. a user study demonstrating that end-users communicate their intended program more efficiently with pragmatic synthesizers

## 2 Program Synthesis as a Reference Game

We now formally connect program synthesis with pragmatic communication. We describe *reference game*, a class of cooperative 2-player games from the linguistic literature. We then cast program synthesis as an instance of a reference game played between a human speaker and a machine listener.

### 2.1 Program Synthesis

In program synthesis, one would like to obtain a program without explicitly coding for it. Instead, the user describes desirable properties of the program as a specification, which often takes in the form of a set of examples. Given these examples, the synthesizer would search for a program that satisfies these examples. In an interactive setting [18], rather than giving these examples all at once, the user gives the examples in rounds, based on the synthesizer's feedback each round.

### 2.2 Reference Game

In a reference game, a speaker-listener pair $(S, L)$ cooperatively communicate a **concept** $h \in H$ using some atomic **utterances** $u \in U$. Given a concept $h$, the speaker $S$ chooses a **set of utterances** $D = \{u^1, \ldots, u^k | u^i \in U\}$ to describe the concept. The communication is successful if the original concept is recovered by the listener, i.e. $h = L(S(h))$. The communication is efficient if $|D|$ is small.

Therefore, it should be unsurprising that, given a reference game, a human speaker-listener pair would act *pragmatically* [3]: The speaker is choosing didactic utterances that are most descriptive yet parsimonious to describe the concept, and the listener is aware that the speaker is being didactic while recovering the intended concept.

## 2.3 Program Synthesis as a Reference Game

It is easy to see why program synthesis is an instance of a reference game: The user would like to obtain a "concept" in the form of a "program", the user does so by using "utterances" in the form of "examples". See Figure 1. This formulation can explain in part the frustration of using a traditional synthesizer, or machine in general. Because while the user naturally assumes pragmatic communication, and selects the examples didacticly, the machine/synthesizer is not pragmatic, letting the carefully selected examples fall on deaf ears.

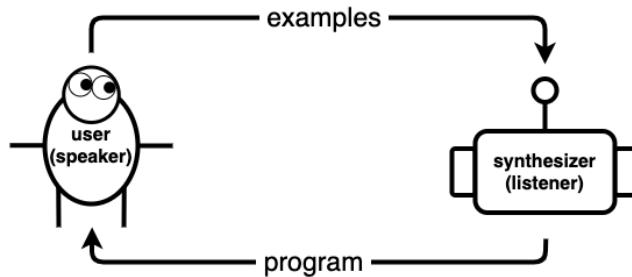

Figure 1: program synthesis as a reference game

## 2.4 Obtaining Conventions in Human-Machine Communication

Two strangers who speak different languages would not perform as well in a reference game as two close friends. Clearly, there needs to be a convention shared between the speaker and the listener for effective communication to occur. Approaches such as [19, 20] use a corpus of human annotated data so that the machine can imitate the conventions of human communication directly. Works such as [21, 22] leverage both annotated data and pragmatic inference to achieve successful human-machine communication over natural language. This work shows that, in the context of program synthesis by examples, by building the concept of pragmatic communication into the synthesizer, the user can quickly adopt the convention of the synthesizer effectively via human learning [2]. This is advantageous because annotated user data is expensive to obtain. In this regard, our work is most similar to SHRDLURN [23], where a pragmatic semantic parser was able to translate natural language utterances into a desirable program without being trained first on human annotated data.

# 3 Communicating Concepts with Pragmatics

We now describe how to operationalize pragmatics using a small, program-like reference game, where by-hand calculation is feasible. This exposition adapts formalism from [18] for efficient implementation within program synthesizers.

**The Game.** Consider the following game. There are ten different **concepts** $H = \{h_0 \ldots h_9\}$ and eight **atomic examples** $\{u_0 \ldots u_7\}$. Each concept is a contiguous line segment on a horizontal grid of 4 cells, and each atomic example indicates whether a particular cell is occupied by the segment. One can view this example as an instance of predicate synthesis, where the program takes in the form of a predicate function $h$, and the atomic examples as input-output pairs obtained by applying the predicate function on some input: i.e. $u_0 = (cell_0, h(cell_0) = True)$. We can visualise the game with a **meaning matrix** (Figure 2), where each entry $(i, j)$ denotes whether $h_j \vdash u_i$ ($h_j$ is consistent with $u_i$). Given a set of examples $D$, we say $h \vdash D$ if $\forall u \in D, h \vdash u$.

If a human speaker uses the set of examples $D = \{u_2, u_4\}$, what is the most likely concept being communicated? We should expect it is $h_5$, as $u_2$ and $u_4$ marks the end-points of the segment, despite

|    |      | h0 | h1 | h2 | h3 | h4 | h5 | h6 | h7 | h8 | h9 |
|----|------|----|----|----|----|----|----|----|----|----|----|
| u0 | ✔    | 1  | 1  | 1  | 1  | 0  | 0  | 0  | 0  | 0  | 0  |
| u1 | ✘    | 0  | 0  | 0  | 0  | 1  | 1  | 1  | 1  | 1  | 1  |
| u2 | ✔    | 0  | 1  | 1  | 1  | 1  | 1  | 1  | 0  | 0  | 0  |
| u3 | ✘    | 1  | 0  | 0  | 0  | 0  | 0  | 0  | 1  | 1  | 1  |
| u4 | ✔    | 0  | 0  | 1  | 1  | 0  | 1  | 1  | 1  | 1  | 0  |
| u5 | ✘    | 1  | 1  | 0  | 0  | 1  | 0  | 0  | 0  | 0  | 1  |
| u6 | ✔    | 0  | 0  | 0  | 1  | 0  | 0  | 1  | 0  | 1  | 1  |
| u7 | ✘    | 1  | 1  | 1  | 0  | 1  | 1  | 0  | 1  | 0  | 0  |

Figure 2: the meaning matrix: entry $(i, j)$ denotes if example $u_i$ is true given concept $h_j$.

the concepts $h_2, h_3, h_6$ are also consistent with $D$. We now demonstrate an incremental pragmatic model that can capture this behaviour with recursive Bayesian inference.

### 3.1 Communication with Incremental Pragmatics

The recursive pragmatic model derives a probabilistic speaker $S_1$ and listener $L_1$ pair given a meaning matrix, and the resulting convention of the communicating pair $S_1$-$L_1$ is shown to be both efficient and human usable [24]. Clearly, there are other ways to derive a speaker-listener pair that are highly efficient, for instance, training a pair of cooperative agents in a RL setting [25]. However, agents trained this way tends to deviate from how a human would communicate, essentially coming up with a highly efficient yet obfuscated communication codes that are not human understandable.

**Literal Listener $L_0$.** We start by building the literal listener $L_0$ from the meaning matrix. Upon receiving a set of examples $D$, $L_0$ samples uniformly from the set of consistent concepts:

$$P_{L_0}(h|D) \propto \mathbb{1}(h \vdash D), \quad P_{L_0}(h|D) = \frac{\mathbb{1}(h \vdash D)}{\sum_{h' \in H} \mathbb{1}(h' \vdash D)} \quad (1)$$

Applying to our example in Figure 2, we see that $P_{L_0}(h_5|u_2, u_4) = \frac{1}{4}$.

**Incrementally Pragmatic Speaker $S_1$.** We now build a pragmatic speaker $S_1$ recursively from $L_0$. Here, rather than treating $D$ as an unordered set, we view it as an ordered *sequence* of examples, and models the speaker's generation of $D$ incrementally, similar to autoregressive sequence generation in language modeling [26]. Let $D = u^1 \ldots u^k$, then:

$$P_{S_1}(D|h) = P_{S_1}(u_1, \ldots, u_k|h) = P_S(u_1|h)P_S(u_2|h, u_1) \ldots P(u_k|h, u_1 \ldots u_{k-1}) \quad (2)$$

where the incremental probability $P_S(u_i|h, u_1, \ldots, u_{i-1})$ is defined recursively with $L_0$:

$$P_S(u_i|h, u_{1\ldots i-1}) \propto P_{L_0}(h|u_{1\ldots i}), \quad P_S(u_i|h, u_{1\ldots i-1}) = \frac{P_{L_0}(h|u_1, \ldots, u_i)}{\sum_{u_i'} P_{L_0}(h|u_1, \ldots, u_i')} \quad (3)$$

Applying this reasoning to our example in Figure 2, we see that $P_{S_1}(u_2, u_4|h_5)$ is:

$$P_S(u_2|h_5)P_S(u_4|h_5, u_2) = \frac{P_{L_0}(h_5|u_2)}{\sum_{u'} P_{L_0}(h_5|u')} \frac{P_{L_0}(h_5|u_2, u_4)}{\sum_{u''} P_{L_0}(h_5|u_2, u'')} = 0.25 * 0.3 = 0.075 \quad (4)$$

**Informative Listener $L_1$.** Finally, we construct an informative listener $L_1$ which recursively reasons about the informative speaker $S_1$:

$$P_{L_1}(h|D) \propto P_{S_1}(D|h), \quad P_{L_1}(h|D) = \frac{P_{S_1}(D|h)}{\sum_{h'} P_{S_1}(D|h')} \quad (5)$$

In our example, $P_{L_1}(h_5|u_{2,4}) \approx 0.31, P_{L_1}(h_2|u_{2,4}) \approx 0.28, P_{L_1}(h_3|u_{2,4}) \approx 0.19, P_{L_1}(h_6|u_{2,4}) \approx 0.21$. As we can see, the intended concept $h_5$ is ranked first, in contrast to the uninformative listener $L_0$.

# 4 Efficient Computation of Incremental Pragmatics for Synthesis

Computing the pragmatic listener $L_1$ naively would incur a cost of $O(|H|^2|U||D|^2)$, which can be prohibitively expensive even in instances where $H$ and $U$ can be enumerated. Here, we give an efficient implementation of $S_1$ and $L_1$ that is drastically faster than the naive implementation. While our algorithm cannot yet scale to the regime of state-of-the-art program synthesizers – where $H$ and $U$ cannot be enumerated – we believe computational principles elucidated here could pave the way for pragmatic synthesizers over combinatorially large program spaces, particularly with when this combinatorial space is manipulated with version space algebras (VSA), as in [2, 5, 17]. To this end, we employ VSA with aggressive precomputation to memoize the cost of pragmatic inference.

## 4.1 Formulation

We start by redefining some terms of pragmatics into the language of program synthesis. Let $h$ be a **program** and $H$ be the **set of programs**. Let $X$ be the **domain** of the program and $Y$ be the **range** of the program: $H : X \to Y$. An **example** $u$ is a pair $u = (x, y) \in X \times Y = U$. A program is **consistent** with an example, $h \vdash u$, if $u = (x, y)$, $h(x) = y$.

## 4.2 Precomputations

We use a simple form of version space algebra [17] to precompute and cache two kinds of mappings. First, we iterate over the rows of the meaning matrix and store, for each atomic example $u$, the set of programs that are consistent with it: $M_L[u] = \{h | h \vdash u\}$. Here $M_L$ is a map or a dictionary data structure, which can be thought of as an atomic listener, that returns a set of consistent programs for every atomic example. Second, we iterate over the columns of meaning matrix, and store, for each program $h$, the set of atomic examples that are consistent with it $M_S[h] = \{u | h \vdash u\}$. $M_S$ can be thought of as an atomic speaker, that returns a set of usable atomic examples for every program. Abusing notation slightly, let's define: $|M_L| = max_u |M_L[u]|$ and $|M_S| = max_h |M_S[h]|$. Note that these quantities can be significantly smaller than $H$ and $U$ if the meaning matrix is sparse.

## 4.3 Computing $P_{L_0}$

To compute $P_{L_0}(h|D)$, we first compute the set intersection $D[H] = \cap_{u \in D} M_L[u]$, which corresponds to the set of programs consistent under $D$. Note $D[H] = \{\} \iff h \nvdash D$. Therefore, from Eq. 1 we derive $P_{L_0}(h|D) = 0$ if $D[H] = \{\}$, and $\frac{1}{|D[H]|}$ otherwise. Each call is time $O(|M_L||D|)$.

## 4.4 Computing $P_{S_1}$

Computing $P_{S_1}$ amounts to computing a sequence of the incremental probability $P_S$ defined in Eq. 3. The brunt of computing $P_S$ lies in the normalisation constant, $\sum_{u_i'} P_{L_0}(h|u_1, \ldots, u_i')$. We speed up this computation in two ways: First, we note that if $h \nvdash u_i'$, the probability $P_{L_0}(h|u_1, \ldots, u_i')$ would be 0. Thus, we can simplify this summation using the atomic speaker $M_S[h]$ like so: $\sum_{u_i'} P_{L_0}(h|u_1, \ldots, u_i') = \sum_{u_i' \in M_S[h]} P_{L_0}(h|u_1, \ldots, u_i')$, which reduces the number of terms within the summation from $O(|U|)$ to $O(|M_S|)$. Second, recall that computing $P_{L_0}(h|D)$ amounts to computing the consistent set $D[H]$. We note that the only varying example inside the summation is $u_i'$, while all the previous examples $D_{prev} = \{u_1 \ldots u_{i-1}\}$ remains constant. This allows caching the intermediate results of the set intersection $D_{prev}[H] = \cap_{u \in D_{prev}} M_L[u]$ to be re-used in computing $(D_{prev} \cup \{u_i'\})[H] = M_L[u'] \cap D_{prev}[H]$, up to $|D|$ times. Thus, $P_{S_1}$ is $O(|M_L||M_S||D|^2)$.

## 4.5 Computing $P_{L_1}$

Again, the brunt of the computation lies in the normalisation constant $\sum_{h'} P_{S_1}(D|h')$ of Eq 5. However, note that in case $h' \nvdash D$, $P_{S_1}(D|h') = 0$. This would allow us to leverage the consistent set $D[H]$ to sum over almost $|M_L|$ elements: $\sum_{h'} P_{S_1}(D|h') = \sum_{h' \in D[H]} P_{S_1}(D|h')$. Overall, $P_{L_1}$ is $O(|M_L|^2|M_S||D|^2)$ time, significantly faster than the original $O(|H|^2|U||D|^2)$.

```
P  -> if (x,y) in box(B,B,B,B)
       then symbol(S,C)
       else pebble
B  -> 0 | 1 | 2 | 3 | 4 | 5 | 6
S  -> ring(O,I,R,x,y)
O  -> chicken | pig
I  -> chicken | pig | pebble
R  -> 1 | 2 | 3
C  -> [red, green, blue][A2(A1)]
A1 -> x | y | x+y
A2 -> lambda z:0 | lambda z:1 |
      lambda z:2 | lambda z:z%2 |
      lambda z:z%2+1 |
      lambda z:2*(z%2)
```

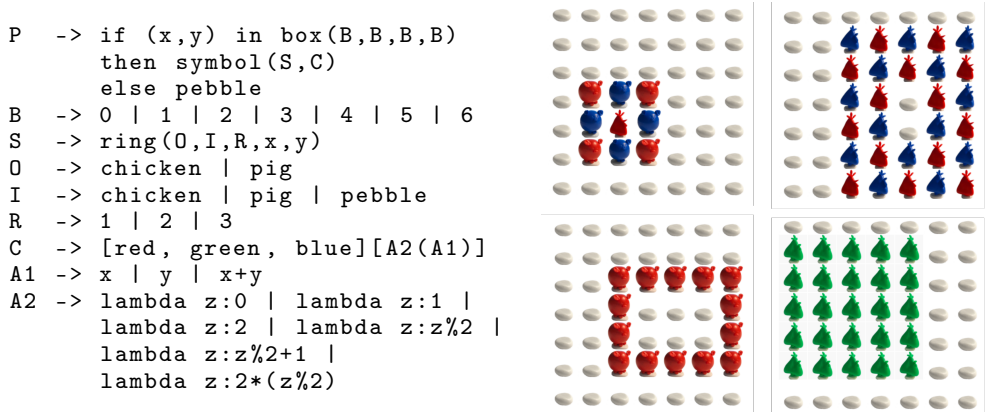

Figure 3: DSL of pattern laying programs / rendering of 4 different programs on $7 \times 7$ grids

## 5 A Program Synthesis System with Pragmatics

To describe our program synthesis system with pragmatics, we only need to specify the space of programs, the space of atomic examples, and the meaning matrix; the rest will follow.[3]

**Programs.** We consider a simple domain of programs that can layout grid-like patterns like those studied in [27,28]. Specifically, each program is a function that takes in a coordinate $(x, y)$ of a $7 \times 7$ grid, and place a particular symbol at that location. Symbols can be one of three shapes: *chicken*, *pig*, *pebble*, and be one of three colors: *red*, *green*, *blue*, with the exception that *pebble* is always colorless. A DSL and some of the programs renderings are shown in Figure 3. Here, *box* is the bounding box where the main pattern should be placed. *ring* is a function that takes two shapes and makes the outside shape $O$ wrap around the inside shape $I$ with a thickness of $R$. *symbol* is a function that takes in a shape and a color and outputs an appropriate symbol. We consider two programs $h_1$ and $h_2$ equivalent if they render to the same pattern over a $7 \times 7$ grid. After such de-duplication, there are a total of 17976 programs in our space of programs.

**Atomic Examples.** The space of atomic examples consists of tuples of form $((x, y), s)$, where $(x, y)$ is a grid coordinate, and $s$ is a symbol. As there are a total of 7 distinct symbols and the grid is $7 \times 7$, there are a total of 343 atomic examples in our domain.

**Meaning Matrix.** An entry of the meaning matrix denotes whether a program, once rendered onto the grid, would be consistent with an atomic example. For instance, let the upper-left pattern in Figure 3 be rendered from program $h_{1337}$, then, it will be consistent with the atomic examples $((0, 0), pebble)$ and $((3, 3), pig\_red)$, while be inconsistent with $((6, 6), pig\_blue)$.

## 6 Human Studies

We conduct an user study to evaluate how well a naive end-user interacts with a pragmatic program synthesizer ($L_1$) versus a non-pragmatic one ($L_0$). We hypothesized that to the extent that the pragmatic models capture computational principles of communication, humans should be able to communicate with them efficiently and intuitively, even if the form of communication is new to them.

### 6.1 Methods

**Subjects.** Subjects (N = 55) were recruited on Amazon Mechanical Turk and paid $2.75 for 20 minutes. Subjects gave informed consent. Seven responses were omitted for failing to answer an instruction quiz. The remaining subjects (N=48) (26 M, 22 F), (Age = 40.9 +/- 12.1 (mean/SD)) were included. The study was approved by our institution's Institutional Review Board.

**Stimuli.** Stimuli were 10 representative renderings of program sampled from the DSL, capturing different concepts such as stripes vs checkered colour patterns and solid vs hollow ring shapes.

**The communication task.** The subjects were told they are communicating with two robots, either white ($L_0$) or blue ($L_1$). The subjects were given a stimuli (a rendering), and were asked to make a robot recreate this pattern by providing the robots with few, strategically placed symbols on a scratch grid (set of examples). Each time the subject places a symbol, the robot guesses the most likely program given the examples, and display its guess as a rendering as feedback to the subject. The subject may proceed to the next task if the pattern is successfully recreated. See Figure 6.1 [4].

**Procedure.** First, the subjects read the instructions followed by a quiz. Subjects who failed the quiz twice proceeded with the experiment, but their responses were omitted. Next, the subjects practice with selecting and placing symbols. Subjects proceed with the communication task presented in two blocks, one with white robot $L_0$ and one with blue robot $L_1$, in random order between subjects. Each block contains 10 trials of the 10 stimuli, also in random order. In the end of the experiment subjects fill a survey: which robot was easier, and free-form feedback about their communication strategies.

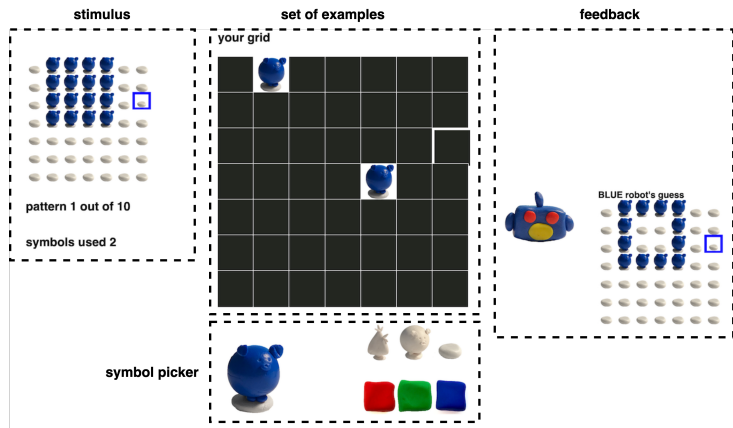

Figure 4: user interface for the communication task

## 6.2   Results

**Behaviour Analysis.** We first compared the mean number of symbols subjects used to communicate with each robot. A paired t-test was significant ($t = 12.877, df = 47, p < .0001$), with a mean difference of 2.8 moves, and a 95% confidence interval $(2.35, 3.22)$. The numbers of symbols used for both robots by subjects is shown in Figure 5 (a).

A linear regression model for the mean number of $symbols$ used as a dependent variable, and $robot$, $trial$ as independent variables, was significant (adjusted $R^2 = 0.95, p < .0001, F(3, 16) = 134.8$), with significant coefficients for robot ($p < .0001$), and trial ($p < .0001$). The regression equation is given by: $symbols = 6.1 + 2.23 * robot - 0.14 * trial + 0.1 * (robot : trial)$, where robot = $\{0 - blue, 1 - white\}$, and trial is the order in which the stimulus was shown to subjects. This concludes that subjects' communication with robots became more efficient over time. The interaction between the variables was close to being significant ($p < .07$), suggests that this communication improvement *might* have been driven by the pragmatic listener (blue robot) (Figure 5 (b)).

A significant majority of subjects (77%, $\chi^2 = 26.042, p < .0001, df = 1$) reported that the blue(L1) robot was easier. This was true regardless of which robot they saw first (Figure 5 (c)).

**Communication Efficiency Analysis.** Next, we compare communication efficiency between different speaker-listener pairs. We consider 3 speakers: S0 (a random speaker that uses any consistent examples, as a lower bound), S1 (the pragmatic speaker that L1 was expecting, as an upper bound),

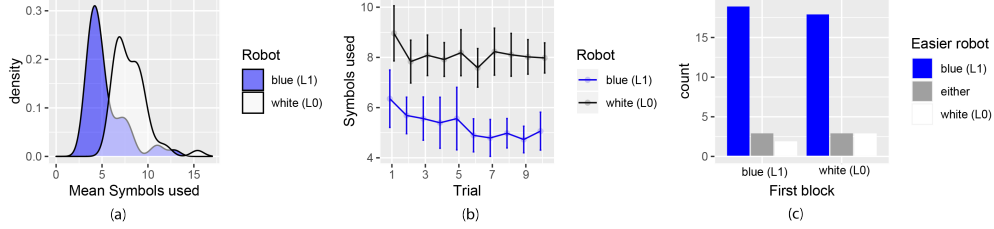

Figure 5: (a) the density of the mean number of symbols used (N=48). (b) the mean number of symbols used by subjects during the course of the experiment (error bars show 95% confidence intervals), communicating with the both robots improvement over time. (c) which robot was easier to communicate with.

and *human*. We consider two listeners: L0 and L1. We first measure the probability of successful communication, $P(L(S(h)) = h)$, as a function of numbers of symbols used by sampling[5] from the speaker and listener distributions (Figure 6 (a)). We find that both human and S1 communicate better with an informative listener L1 rather than L0. We then measure the mean number of symbols required for successful communication between a speaker-listener pair[6] (Figure 6 (b)). A one-way ANOVA testing the effect of speaker-listener pair on number of symbols used was significant ($F(4, 45) = 66, p < .0001$), with significant multiple comparisons between means given by Tukey test for the following pairs: S0-L0 vs human-L0 ($p < .0001, d = 8.4$), S1-L0 vs human-L0 ($p = .004, d = 3.5$) and human-L0 vs human-L1 ($p = .3, d = 2.8$). There were no significant differences between S1-L1 vs human-L1 ($p = .2$) and between S1-L1 vs S1-L0 ($p = .6$). This means that human communication is significantly more efficient compared to the uninformative speaker (S0), and for the pragmatic listener, human efficiency is indistinguishable from the pragmatic speaker (S1). Further, compared to the pragmatic model S1, humans were significantly less efficient when communicating with the literal listener L0. This suggests that humans intuitively assume that a listener is pragmatic, and find communication difficult when this assumption is violated. This may have implications when engineering systems that do few-shot learning from human demonstration.

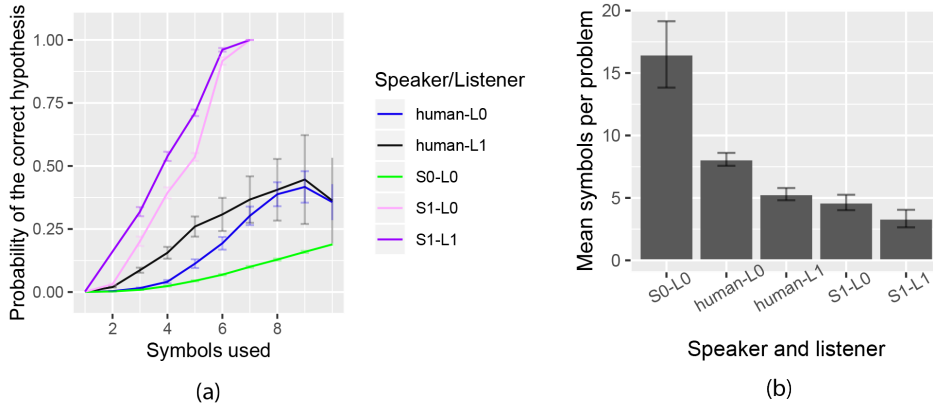

Figure 6: (a) probability of successful communication as a function of symbols used (up to 10). (b) mean number of moves for speaker-listener pair, error bars indicate 95% confidence intervals.

**Comparison Against a Crafted Prior** We conduct an experiment comparing the pragmatic listener $L_1$ against a listener $L_p$ that leverages a prior to disambiguate the programs – a common strategy in previous works. Formally, $P_{L_p}(prog|D) \propto \mathbb{1}[prog \vdash D] P(prog)$, where $prog \vdash D$ means $prog$ is consistent with spec $D$, and $P(prog)$ is a prior over programs. Given a pattern obtained by executing program $prog$, let $sym(prog)$ be the number of non-pebble symbols in the pattern, and let $kinds(prog)$ counts the distinct kinds of non-pebble symbols in the pattern. For example, the

upper-right pattern in Fig 3 has $sym(prog) = 28$ and $kinds(prog) = 2$. We craft the following prior: $P(prog) \propto 100sym(prog) + kinds(prog)$. This prior will firstly prefer patterns with fewer non-pebble symbols, and secondly prefer patterns with fewer kinds of symbols.

Using $S_1$ as the speaker, we compare the mean number of utterances required when $L_1$ is the listener against $L_p$. The S1-L1 pair (mean 3.34, std 1.07) has slightly better performance to the S1-Lp pair (mean 3.8, std 1.08). This is encouraging, as $L_1$ is derived from the meaning matrix and the principles of pragmatic communication alone, without any hand-crafting [4] or training on real-world data [29].

**User Adopted Conventions**    Here we select a few adopted conventions of end-users as they communicate with the two listeners. These were collected in the exist survey of our study.

| adopted conventions for communicating with L0 | adopted conventions for communicating with L1 |
| --- | --- |
| I had to provide a lot more objects, and it seemed necessary to use the pebbles for it to narrow down the field to the correct size. | It wasn't necessary to provide pebbles on the outside like it was with white and it would know the size based on the opposite corners . . . |
| He needed more help than the blue robot so it just took longer to get him to the point. you needed to tell him about more of the while spaces | The blue robot was easy i would just lay out the boundary and he would get it fast most of the time. |
| Try to place a piece on a corner somewhere so let it find a boundary, then work to correct its errors. | Honestly it was pretty hit-and-miss and I couldn't easily figure out what it was doing. |
| I put the figures in a way to outline the "box" - the white robot needed to have things spelled out more | Same as above, only the blue robot made inferences better. |
| Try to show the dimensions and what type of piece goes where. | Try to pick the most different pieces and place at the outside most area of where they appear. |
| I tired to use the white pebbles more to block areas off | The blue robot seemed to figure it out pretty quickly with diagonally placed figures. |

Figure 7: user adopted conventions

Overall, for the non-pragmatic robot, the users needed to use more pebbles to limit the pattern's size, whereas the pragmatic robot can intuitively infer the size of the pattern if they showed it corners.

# 7   Looking Forward

In this work, we show that it is possible to obtain a pragmatic program synthesis system by building the principles of pragmatic communication into the synthesis algorithm rather than having it train on actual human interaction data. However, a system that can adapt online to the user would be even more valuable. It is also interesting to see whether version space algebra approaches would scale to more complex program synthesis domains. Approximating pragmatic computation have been explored in [21, 30] where the number of hypotheses is small. It would be interesting to see if these approaches can be adapted to work over a combinatorially complex hypothesis space of programs. In general, we believe interactive learning systems are a prime target of future research: not only do we desire machines that learn from massive data, but also machine intelligence which can acquire knowledge from pedagogy and communication.

## Broader Impact

We hope that naive end-users would benefit from this research, as we aim for a more natural interaction between human and machine. This would allow boarder access to computes by non-programmers, so that we may work along-side the machines rather than being replaced by them.

## Acknowledgement

Thanks Maxwell Nye for drawing the glasses and hat figure on the board and introducing me to the wonderful world of pragmatics. Thanks MH Tessler for explaining RSA to me in detail. Thanks Beilei Ren for making the clay figures used in the user study, and designing the page layout. Thanks twitch chat for support POG. Funded by the National Science Foundation under Grant No. 1918839.

## Footnotes

[2]which is far more powerful than machine learning

[3]code : https://github.com/evanthebouncy/program_synthesis_pragmatics

[4]play with the sandbox mode here! : `https://evanthebouncy.github.io/projects/grids/`

[5]instead of picking the top-1 program

[6]taking the top-1 program from the listeners instead of sampling

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
