[Reviews · NeurIPS 2020]

Review 1

Summary and Contributions: This paper frames interactive program synthesis as a reference game played between a demonstrator and a synthesizer. The setting is constructing patterns on a 2D grid, where the demonstration iteratively constructs a pattern by placing symbols, and the synthesizer infers a program to complete the output based on the symbols produced so far. The paper applies recursive pragmatic models from the rational speech acts (RSA) framework, deriving a pragmatic synthesizer that models the demonstrator's intention in choosing symbols. This is done by alternating renormalization over demonstrations and over programs, using full enumeration of the set of possible programs and memoization of probabilities. The paper compares pragmatic and non-pragmatic synthesizers, with humans playing the role of the demonstrators. Pragmatic synthesizers have significantly more efficient interactions: the human needs to place fewer symbols on average in order to correctly get the synthesizer to infer the pattern the person was attempting to demonstrate. --- update after response --- Thanks to the authors for the thorough response, which clarified nearly all of my questions and also partly addressed the weaknesses I saw. After this, and considering the other reviews, I've raised my score to a 7 (from a 6). Thanks for the comments on scalability. I think that the large output space is interesting (especially when combined with the incremental pragmatics). I agree this work could lead to future work in more complex program domains, using either of the proposed approaches. Amortizing L_1 is likely to be tricky, but learned parameterizations for L_0 and S_0, e.g. following Wang et al., do seem promising. The comparison to a prior is informative, thank you for doing this. It would be helpful to also include more details about how the prior is defined given space in any future version. It would be interesting to see if the pragmatic listener further improved if it incorporated this prior. I also found the example about how incremental pragmatics can capture order-dependent distributions, interesting and informative. Minor point: - I'm a bit unclear on argmax vs sampling in the efficiency results. Why is the soft score desirable? It might be helpful to be consistent between Figure 5 and 6, or present the results for both, or motivate the inconsistency more clearly.

Strengths: I found the framing of program synthesis as a communication game well-motivated and (to my knowledge) underexplored. The paper suggests interesting future work on using similar approaches in conjunction with learned models, or in domains with more complex programs. The human subject experiments and analysis seemed thorough, and showed a significant effect of the pragmatic reasoning procedure on efficiency.

Weaknesses: If I understood correctly, by removing redundant programs (that execute to the same pattern), the approach doesn't deal with the structure of the programs, and the setup effectively reduces to a referring expression setup of a type well-studied in other literature on RSA (e.g. Cohn-Gordon et al. 2019), albeit with a very large space of possible references. The paper would be stronger if it shows that the approach could also be applied in a more complex domain where the space of programs, or their denotations, can't be fully enumerated. The proposed pragmatic synthesizer, L1, is only compared to a weak baseline synthesizer L0 which assigns uniform probability to all consistent programs. The paper would be stronger if it compared to a method that defined a prior over reasonable programs. Alternatively, it would help to provide some analysis of how large the space of consistent programs is at each timestep. --- update after response --- Both of these weaknesses were addressed in the author response (thanks!). The domain here is enumerable but still large; the response proposes some viable approaches for making this scale to non-enumerable domains. The response also presents analysis using a prior.

Correctness: The method seemed correct and in line with past work on RSA and computational pragmatics, and I didn't see any issues with the methodology and evaluation. I felt there was currently too much emphasis on the claim in lines 103-106: "This work shows that... by building the concept of pragmatic communication into the synthesizer, the user can quickly adapt to communicate with the synthesizer effectively via human learning." Is this supported by more than the interaction effect between trial number and literal vs pragmatic listener on efficiency mentioned in 241-243, which was small? Line 262 suggests that "humans intuitively assume a listener is pragmatic, and find communication difficult when this assumption is violated". While this seems plausible, an alternative explanation could be that the L0 listener is just worse at making predictions (i.e. if the L1--human interactions were replayed with the L0 listener, the performance would be similar to the interactions collected with the L0 listener).

Clarity: The paper was clear overall, particularly in the motivation for the approach and the description of the RSA pragmatic framework. It would help to spend more time describing how version space algebra is used in the approach -- is this just the caching and set intersection described briefly in 4.4 and 4.5, or something more?

Relation to Prior Work: Wang et al. 2016 [23] also explore using pragmatics for program synthesis, although in several ways their setting is more complex -- using base models that learn from data in an online fashion, with approximate inference over the space of programs, and language demonstrations. It would be helpful to describe what this work adds to the work of Wang et al.

Reproducibility: Yes

Additional Feedback: - 149: "that runs in worst polynomial of |H| |U| time" : this is unclear, does it mean polynomial in |H| |U|? or a constant times |H||U|, which is polynomial? - 203: I had difficulty matching the example given with Figure 3 -- which pattern is referred to by "left-most", and what does program h_1337 refer to? - 242: "which was small but not significant": this wording is odd - It wasn't clear to me why, for the efficiency experiments, accuracies are computed by sampling from the listeners L0 and L1 rather than taking the argmax. If I understand correctly, since L0 has a uniform distribution over consistent utterances, sampling would be equivalent to an argmax with random tiebreaking -- but would the performance of L1 improve if taking the argmax? - It would help to give some detail on whether the user can remove a demonstration, and if so how this is incorporated into the incremental pragmatics model. - Did all of the human subject trials result in success? If not, what is the difference in success rates between the L0 and L1 conditions? The broader impacts section could use revision: it was unclear how to me how the approach presented here dealt with dissection of network architecture, and I think more detail would be needed to support the point that it can "be more complicated to prove and verify whether an AI system is working as intended in a complex communication setting". Minor points: - Using sentence fragments as footnotes felt strange stylistically. - 276: "we aim for a more natural interaction between human and machine" (grammar)


Review 2

Summary and Contributions: The paper presents a pragmatic inference perspective on program synthesis. A program synthesizer typically takes as input a specification given in the form of input-output examples. These examples are constructed by humans, and often with the explicit intention that others looking at them can correctly identify an underlying program. The method proposed in this work explicitly models the input-output examples under a pragmatic communication theory, where the synthesizer takes into account the fact that the speaker is carefully choosing examples for the specification. Results show that the proposed pragmatic synthesizer is more efficient when communicating with humans in a simple example domain.

Strengths: The paper's novelty is largely in presenting program synthesis from the perspective of pragmatic communication. This idea seems very interesting as it provides a method for constructing a program prior that aligns well with that of humans. Beyond driving the human-computer interfaces of program synthesizers, this prior might of interest to the field more generally, e.g. by improving generalization in deep program synthesizers, or when generating non-human-obtained data to train a program synthesizer, where the pragmatic communication perspective can be a principled method to sample input-output examples aligned to the distribution of what a human would choose. The paper's results have significance for the field, as in the human interaction study presented it empirically confirms the hypothesis that humans have a preference for a pragmatic synthesizer over a literal one, motivating this research direction as being promising for human interface design of program synthesis and a general prior for programs.

Weaknesses: The most significant weakness with this work is that it requires explicit enumeration of the meaning matrix, which even in smaller applications seems like it could grow intractable very quickly. Although the authors presented efficiency improvements through memoization, I am not sure how well these alone can cause the synthesizer to scale to comparable program domains as state-of-the-art ones currently do. Therefore from my understanding the method is currently limited to smaller program domains or ones where a significant engineering effort is spent on constructing the program space. However, I think the work does a good job of motivating interest in this direction and could spur future research into faster or approximate methods for pragmatic program synthesis.

Correctness: I did not find any incorrect claims in the paper and the empirical methodology seems sound.

Clarity: The paper is well written and is largely self-contained. I found the method exposition in section 3 easy to follow and the review on pragmatics in section 2 helpful.

Relation to Prior Work: The paper clearly discusses how it is related to previous work. The method of pragmatic program synthesis from input-output examples is a novel contribution as far as I can tell, and they mention a closely related work that looked at pragmatic program synthesis from natural language.

Reproducibility: Yes

Additional Feedback: Post-Rebuttal: Thank you for the detailed rebuttal. I maintain my vote for acceptance.


Review 3

Summary and Contributions: When learning programs from examples (program synthesis), potentially many programs are consistent with the examples provided to the learner. The authors of this paper propose a novel approach to disambiguate between consistent programs based on the principles of pragmatic communication. Their approach allows humans to communicate with a computer more effectively.

Strengths: - the idea seems both significant, novel and relevant -- definitely interesting! - while there are few theoretical claims in the paper they seem sound

Weaknesses: W1 The major weakness of the paper is the lack of technical contributions as to how the proposed approach can be efficiently implemented in a program synthesis tool (the authors acknowledge this shortcoming). W2 The discussion to related work is somewhat shallow and lacks a more detailed explanation of how and in what cases the proposed approach disambiguates differently from other approaches. Such other approaches are also not compared to in the evaluation which is limited to comparing to versions proposed by the author.

Correctness: The math and general claims seem correct, however, I am unfamiliar with the field of linguistics and, therefor, cannot attest to claims as related to that subject.

Clarity: It is fine.

Relation to Prior Work: Not completely, see W2.

Reproducibility: Yes

Additional Feedback: Weakness W1, to me, prevents this from being an excellent paper but is also inherent in the scope you chose. Weakness W2 prevents this from being a solid paper, so I hope you can address my concerns in your answer. Therefore, I would ask you if you could provide me with a brief outline of how you could substantiate the comparison to related work? Post rebuttal: The author feedback adequately addressed my main concerns. In combination with the sentiment of the other reviews this makes me retain my score and recommend the acceptance of this paper.


Review 4

Summary and Contributions: This paper uses a rational communication approach to build a probabilistic model that infers a program in a small grid-world maniuplation programming language. The paper also presents a user study showing that humans tend to choose input/output examples consistent with rational communication. Thus, a programming-by-example agent that makes the rational communication assumption requires fewer input/output examples from a human to synthesize the correct program.

Strengths: The domain is certainly of interest to the NeurIPS community, and there are related works that learns to rank possible programs for program synthesis. From what I recall, the related works do not typically have a strong mathematical formulation, and do not have strong user studies. A mathematical formulation of the kind of assumptions that rational agent (or humans users) would make when interacting with a PBE system could be significant.

Weaknesses: The most compelling aspect of the paper is the user study demonstrating that humans subject does, umprompted, assume rational communication. on the part of the synthesizer. This result may be more relevant at a Human-Computer Interaction conference (e.g. CHI) than at NeurIPS. The formulation requires a tractable hypothese (program) and example space, and the authors note that the approach is not scalable. However, the approach could still work well for block-based programming systems and tools that introductory learners use to learn programming. UPDATE: Upon reflection, I think that encouraging work that take into account the human factor in synthesis could be a positive for the NeurIPS community.

Correctness: I may be misunderstanding something, but I'm not sure if the math is correct. In particular, going from equations 3 to 4, it seems that we are missing a few terms. Equation 4 does not condition on the user *not* choosing an example. That is, the conditioning is not on the $u_0$, $u_1$, ... Is this intentional? Are both formulations equivalent? UPDATE: From the author feedback it seems like I did misunderstand something.

Clarity: The paper is well structured and well written. Comments: - Line 129, you might want to spell out "RL"="Reinforcement Learning" - The resolution of the figures could be improved

Relation to Prior Work: There are other contributions in the machine learning field like [1] where the agent *asks* the users for additional examples. The formulation is different, but the end goal of solving a PBE problem with few input/output examples is similar. [1] Laich et al. https://openreview.net/pdf?id=BJl07ySKvS

Reproducibility: Yes

Additional Feedback:

[Author Response · NeurIPS 2020]

Thanks for the engaging comments. We will first address two common concerns, then address the reviewers individually.

**Scalability to Larger Domains**   A universal concern is that this work will not scale to bigger synthesis domains. This
is true. However, the focus of this paper is to define the computational problem and showing why solving it is valuable,
which should in turn spur future research on efficient algorithms. For instance, one may imagine a compositional
synthesis regime where the system communicates with the user to specify a component at a time (for instance, the layout
domain maybe factorized in such a way that the user first specifies the shape patterns, then the color patterns). This way,
rather than enumerating the entire space of programs, we enumerate one component at a time. Another solution is to
amortize each component of the RSA model, $L_0, S_1, L_1$, with neural networks: A neural program synthesizer can serve
as $L_0$. For natural language utterances, "Reasoning about Pragmatics with Neural Listeners and Speakers" (Andreas et.
al.) gives a good construction for a pragmatic speaker $S_1$. For input-output based utterances, "Selecting Representative
Examples for Program Synthesis" (Pu et. al.) gives a good construction for a pragmatic example-selector. Then, one
can conceivably construct $L_1$ on top of the neurally approximated $L_0$ and $S_1$.

**Comparison Against a Baseline that Uses a Prior**   Another common concern
is the lack of a direct comparison against stronger baselines, specifically, one that
leverages a good prior to disambiguate programs (a common strategy in previous
works). Such priors can be obtained in 2 ways: by learning from real-world data,
which is often expensive to obtain or unavailable, or by hand-crafting, which
is appropriate for us. For our domain, a reasonable hand-crafted prior is one
that prefers patterns with fewer numbers and kinds of symbols. We use $L_p$ to
denote the listener that first filters for consistent patterns, then ranks them using
this prior. Using $S_1$ as the speaker, we compare the mean number of utterances
required when $L_0, L_p, L_1$ are the listeners (see Fig). We note that $L_1$ and $L_p$
have similar performances, despite $L_1$ being derived from the meaning matrix

number of utterances required for successful
communication, error bars are 1 std.

and the principles of pragmatic communication alone and without any hand-crafting. This is unsurprising, as $S_1$ and
$L_1$ are *constructed* to be good "partners". On the other hand, given a listener $L_p$ that leverages an arbitrary prior, it is
unclear how to obtain a good speaker "partner" for it other than directly optimizing one against it. We hope to add
$L_p$ to our user study to see whether end-users can find $L_p$ as intuitive as $L_1$. Finally, a pragmatic listener can capture
subtle aspects of communication *that no prior can model*: In the line game (Fig 2 of paper), if the utterance sequence is
$u_3, u_5$, the most likely hypothesis according to $L_1$ is $h_9$, however, if the utterance reversed to $u_5, u_3$, the most likely
hypothesis becomes $h_0$ instead. Thus, we have two valid hypothesis, $h_0$ and $h_9$, with their ordering flipped depending
on the utterance. In contrast, a prior can only model a fixed global ordering over valid hypothesis.

**R1**   • "analysis ... consistent programs at each timestep". There is not much difference, it really is the probability of
$L_1$ that gives it the edge over $L_0$, will add to appendix. • "lines 103-106 ... which was small". Sorry, by small we mean
$p < 0.06$, so not quite $p < 0.05$. • "is this just the caching". Yes, and also realizing parts of the summations are always
0. • "adds to the work of Wang et al". The biggest difference would be their work is synthesis via translation, i.e. from
NL instruction to PL execution. This work is synthesis from examples. Also, the number of candidate programs at each
interaction "step" for Wang is $\sim 30$, for us it is $\sim 3000$. • "what is the runtime?". We re-did the math more carefully,
here is the result: Naively it is $|H|^2|U|k^2$, where $k$ is the number of utterances. The efficient implementation is
$M_L^2 M_S k^2$, where $M_L$ is the maximum number of hypothesis that any atomic utterance can be consistent with ($\sim 3000$),
and $M_S$ is the maximum number of atomic utterance that can be consistent with any hypothesis (49). • "referred to by
"left-most"". It should be "upper-left". • "It wasn't clear to me ... argmax?". Yes it would definitely increase. Here we
want to measure a more "soft" score, as Figure 5 already uses random-tie-break for white and argmax for blue. • " user
can remove a demonstration". We observe that subjects are mostly undoing accidents. • "Did all of the human subject
trials result in success?". Subjects who did not complete the experiment were not used in the analysis.

**R2, R3**   Thanks for the encouragement, the two general responses should cover your concerns. We'll use the space
here to give a simple illustrative example why a prior is not enough in modeling some pragmatic behaviours. Imagine
there is a race of 4 people which I attended, and I tell you "Darn I didn't get first place". The most likely outcome is that
I placed 2nd, more likely than if I'm 3rd. However, if I tell you "Hey at least I'm not last". The most likely outcome is
that I placed 3rd, more likely than if I'm 2nd. No prior orderings of 1st,2nd,3rd,4th can model this.

**R4**   • "from equations 3 ... equivalent?". The speaker model is additive: you start with 0 examples, then add 1, then
add another. Equation 4 is the result of applying both equation 2 and 3, in the case that the user used examples u2, then
u4, to describe hypothesis h5. • "field like [1]". It was a great read! Their work amounts to learning a prior: rather than
a prior over the programs directly, it learns a prior over the specs of the programs (i-o pairs) and implicitly ranked the
programs that way. It also requires training using existing ground-truth data, which our work avoids. We will cite them.

[Meta-Review · NeurIPS 2020]

This paper studies the problem of programming by example via the lens of rational communication: how can we synthesize programs assuming humans are providing examples in a rational communcation framework? There are some significant weaknesses in the computational aspects of the paper, where it depends on explicit enumeration that limits its scalability. Having said that, reviewers (and AC) are in agreement that this is an interesting new idea that is worth publishing. I agree with R4's updated assessment that "Upon reflection, I think that encouraging work that take into account the human factor in synthesis could be a positive for the NeurIPS community."